# Scale-Up Strategy in Quality by Design Approach for Pharmaceutical Blending Process with Discrete Element Method Simulation

**DOI:** 10.3390/pharmaceutics11060264

**Published:** 2019-06-06

**Authors:** Su Bin Yeom, Du Hyung Choi

**Affiliations:** Department of Pharmaceutical Engineering, Inje University, Gyeongnam 621-749, Korea; subin1726@hanmail.net

**Keywords:** discrete element method, quality by design, physics-based model, scale-up strategy, critical process parameter, process simulation

## Abstract

An approach combining quality by design (QbD) and the discrete element method (DEM) is proposed to establish an effective scale-up strategy for the blending process of an amlodipine formulation prepared by the direct compression method. Critical process parameters (CPPs) for intermediate critical quality attributes (IQAs) were identified using risk assessment (RA) in the QbD approach. A Box–Behnken design was applied to obtain the operating space for a laboratory-scale. A DEM model was developed by the input parameters for the amlodipine formulation; blending was simulated on a laboratory-scale V-blender (3 L) at optimal settings. The efficacy and reliability of the DEM model was validated through a comparison of simulation and experimental results. Change of operating space was evaluated using the validated DEM model when scaled-up to pilot-scale (10 L). Pilot-scale blending was simulated on a V-blender and double-cone blender at the optimal settings derived from the laboratory-scale operating space. Both pilot-scale simulation results suggest that blending time should be lower than the laboratory-scale optimized blending time to meet target values. These results confirm the change of operating space during the scale-up process. Therefore, this study suggests that a QbD-integrated DEM simulation can be a desirable approach for an effective scale-up strategy.

## 1. Introduction

The importance of quality by design (QbD) for risk management, cost reduction, and meeting regulatory requirements in product and process development has been emphasized in the pharmaceutical industry over the past decade [1]. The fundamental goal of QbD is to develop pharmaceutical products that consistently satisfy patient’s healthcare needs in terms of safety, efficacy, and performance, as defined in the quality target product profile (QTPP) [2]. For this purpose, QbD can be divided into several steps: (a) Critical quality attributes (CQAs), which are directly linked to the safety and efficacy of pharmaceutical products, are adopted in the QTPP through critical assessment, such as prior knowledge and pre-clinical or clinical studies. (b) Risk assessment (RA) is performed to assess the degree of risk for material attributes and process parameters on CQAs. Critical material attributes (CMAs) and critical process parameters (CPPs) are identified. (c) The effects of CMAs and CPPs on CQAs are quantified to identify their relationship. This step can be performed through design of experiments (DoE). (d) A multi-dimensional design space consisting of a combination of CMAs and CPPs ensures that desirable CQAs are derived through the DoE. A control strategy is established based on this design space to ensure the safety, efficacy, and performance of pharmaceutical products [1,2,3,4,5].

Initial research on the development of the pharmaceutical manufacturing process was performed through DoE at the laboratory-scale to establish an efficient control strategy [4,5]. A multi-dimensional design space consisting of a combination of CPPs was generated based on the laboratory-scale experimental data. However, process variability—batch size or manufacturing equipment changes—during the scale-up process makes it difficult to transform the design space to a larger scale (e.g., pilot-scale or commercial scale) [4,6,7]. Therefore, an alternative approach is required for pharmaceutical QbD to establish a control strategy for efficient scale-up of the manufacturing process. Recently, the manufacturing process simulation, performed using a physics-based model, has been applied in the pharmaceutical industry [8,9,10].

Physics-based models are preferred because they can provide a detailed scientific explanation of the manufacturing process [2,9]. In addition, these models are fairly predictable, even outside areas of the experimental space, i.e., they can easily be scaled up, as they are based on the equations of mass, momentum, and energy conservation in the manufacturing processes [10,11]. Physics-based models include the discrete element method (DEM), computational fluid dynamics (CFD), and finite element method (FEM) [2]. DEM is a numerical method that simulates the dynamic behavior of particles in a system [12]. CFD provides insights into the mathematical physics of fluid flow [13]. FEM predicts the behavior of particles by dividing the process into finite elements or volumes. Both models are located in the field of continuum mechanics [8,14,15]. DEM is commonly used to simulate pharmaceutical manufacturing processes, such as blending, granulation, and coating [10,16,17,18].

Blending is an important and common manufacturing process for preparing a solid dosage form of pharmaceutical drugs [19]. Blending is performed mainly in rotating devices, such as V-blenders, double-cone blenders, and cubic blenders [20,21]. The mixture should have a degree of homogeneity during blending to ensure the quality of solid dosage forms, such as tablets and capsules [22,23]. The homogeneity of the mixture is influenced by several factors, such as material attributes (e.g., particle density, shape, size distribution, surface properties, and cohesive strength) and process parameters (e.g., design of blender, rotational speed, filling level, and blending time) [22,24,25]. These factors affect the agglomeration and segregation of the mixture during the blending process, which affect the mixture’s homogeneity. However, experiments are not sufficient to confirm changes in the agglomeration and segregation of the mixture caused by these factors [5]. Blending simulations can provide deeper insights into material attributes, process parameters, and blending mechanisms for the blending process [26]. A DEM simulation might be useful in that it would provide detailed information that could not be obtained through experiments on the blending process [22].

This study develops a DEM model based on the QbD approach for an efficient scale-up strategy of the blending process for amlodipine formulation. Intermediate critical quality attributes (IQAs) and CPPs for the blending process were identified in the QbD approach, and the operating space was generated by performing a laboratory-scale DoE in a V-blender. The DEM model was developed by defining input parameters, such as the material properties and interaction parameters of each material. The developed DEM model with the optimal setting was used to simulate laboratory-scale blending. The results were compared with experimental results obtained under the same conditions to validate the developed DEM model. Changes in operating space that occur during the scale-up process, including batch size and equipment changes, were validated using the DEM model. Pilot-scale blending simulations were performed in a V-blender and double-cone blender, and the results were statistically evaluated with experimental results. The QbD-based DEM model developed in this study can be an efficient alternative approach to the scale-up strategy of the pharmaceutical manufacturing process. In addition, this study can provide the basis for further studies regarding scale-up strategy in other pharmaceutical manufacturing processes.

## 2. Materials and Methods

### 2.1. Materials

Amlodipine besylate was supplied by Daewon Pharmaceutical Co., Ltd. (Seoul, Korea). Silicified microcrystalline cellulose (Prosolv SMCC^®^ 90) and croscarmellose sodium (CCM-Na) were purchased from JRS PHARMA GmbH & Co. KG (Rosenberg, Germany). Polyvinylpyrrolidone (PVP K25) was purchased from BASF AG (Ludwigshafen, Germany). Magnesium stearate (St-Mg) was purchased from Sigma-Aldrich Co. (St. Louis, MO, USA). All other reagents were of analytical or HPLC grade and were used as received.

### 2.2. Application of QbD Approach

#### 2.2.1. Definitions of IQAs and CPPs for the Blending Process

IQAs, which are directly related to the CQAs of a finished drug product, should be within certain limits, ranges, or distributions, to ensure product quality [27]. The IQAs of the mixture corresponding to the intermediate product for the blending process were established based on their association with homogeneity of mixture, which is directly related to the CQAs of the finished drug product [5]. The homogeneity of mixture is expressed by drug content and content uniformity (CU), and it is related to flowability, which is expressed through the Carr index [28]. Therefore, the drug content, CU, and Carr index were adopted as the IQAs of the blending process of the amlodipine formulation.

The CPPs for the blending process affecting the IQAs were established through RA. Failure mode and effects analysis (FMEA) was used to quantify the degree of risk for the blending process parameters with the IQAs. The degree of risk for each process parameter was determined by its severity, probability of occurrence, and detectability. Severity was applied to measure the possible outcomes of a failure mode and presented the effect on product quality. Probability of occurrence indicated the probability of failure, and detectability was defined as the ability to determine the presence of a failure mode. The relative risks represented by each process parameter were ranked according to a risk priority number (RPN) for corrective actions [29]. The RPN was calculated as given below:(1)RPN=Severity (S)× Probability of occurrence (P)× Detectabtility (D).

The severity, probability of occurrence, and detectability levels were given scores of 1, 3, 5, 7, and 9. For example, the scores of severity were ranked based on the degree of impact the process parameter had on the IQAs, from low to high. A score of 1 for severity implies that the process parameter has no impact on the IQAs, while a score of 9 implies that the parameter has a considerable impact on the IQAs. The probability of occurrence was ranked based on the probability of a failure mode for IQAs, with a score of 1 implying that failure will rarely occur and 9 implying that failure is regularly expected to occur. Detectability was scored based on the degree of failure mode detection for the IQAs. A score of 1 is assigned for detectability when failure is almost always detected, and a score of 9 is assigned when failure cannot be detected using available equipment or methods. Based on these scores of severity, probability of occurrence, and detectability, the RPN scores, calculated using Equation (1), ranged from 1 to 729. Risk level was classified as low for RPN scores below 82, medium for RPN scores ranging from 82 to 245, and high for RPN scores exceeding 246. In addition, the Pareto chart was applied based on the RPN scores of the blending process parameters for each IQA. In the Pareto chart, the threshold value, which is the baseline for the RPN score, was calculated by applying 90% confidence based on the maximum RPN score of 729, as given below:(2)Threshold value=729×(1−0.9)=72.9.

The process parameters representing RPN scores above the threshold value should be carefully controlled during the blending process.

#### 2.2.2. DoE for Blending Process

A Box–Behnken design was applied to optimize the laboratory-scale blending process parameters, using Design-Expert^®^ software (version 10; Stat-Ease Inc., Minneapolis, MN, USA). The Box–Behnken design was used with three control factors—filling level (*x_1_*), rotational speed (*x_2_*), and blending time (*x_3_*). The response factors were set as drug content (*y_1_*), CU (*y_2_*), and the Carr index (*y_3_*). The target values of the IQAs were determined based on prior knowledge and experience from published literature or analysis [30,31]. After conducting 15 experiments using the Box–Behnken design, statistical parameters, including the multiple correlation coefficient (*R*^2^), adjusted multiple correlation coefficient (adjusted *R*^2^), and predicted multiple correlation coefficient (predicted *R*^2^), were evaluated to verify the best-fit model for predicting the quantitative effect of control factors on the response factors. The multiple correlation coefficient was the variation in the responses that could be explained by the model; the adjusted multiple correlation coefficient was the variation in the responses that could be explained by the model, adjusted for the number of predictors; the predicted multiple correlation coefficient was how well the model predicted the removed observation. All of these statistical parameters values ranged between 0 and 1, and closer to 1 indicated the goodness of fit of the data to the model. In addition, the control factors were optimized to satisfy the target values of the response factors by using the statistically verified best-fit model. Design spaces consisting of a combination of optimized control factors were suggested by Design-Expert^®^ software. Based on these design spaces, a Monte Carlo simulation was performed on MODDE^®^ software (version 12; Umetrics, Umeå, Sweden), to obtain a robust operating space, excluding potential failure probabilities in the design space. Monte Carlo simulation is a modeling technique that is often used for probabilistic risk assessment by performing a set of iterative simulations using random numbers [32,33]. A total of 10,000 simulations using the random numbers were performed for all derived design spaces, resulting in operating spaces with a failure probability of less than 1% for each design space. The optimal settings for the control factors were confirmed from the generated operating space.

The composition of the amlodipine formulation was as given below: amlodipine besylate (6.935% *w*/*w*), SMCC 90 (77.065% *w*/*w*), PVP K25 (4.5% *w*/*w*), CCM-Na (10% *w*/*w*), and St-Mg (1.5% *w*/*w*), based on the equivalent amount of one tablet (100% *w*/*w*). The mass of one tablet was 100 mg. All raw materials for the amlodipine formulation were passed through a sieve (#25 mesh) to prevent the material from aggregating before the blending process. Based on the designed filling level, all the materials were mixed in the V-blender (Yenchen machinery Co., Ltd., Taoyuan, Taiwan) at laboratory-scale (3 L). The total volumes of all the materials for the formulation added in the V-blender at 30%, 50%, and 70% filling levels were equal to 0.9 L, 1.5 L, and 2.1 L, respectively. Then, the V-blender was rotated for a designed blending time at a designed rotational speed. The mixtures blended during this time were evaluated for drug content, CU, and the Carr index.

Pilot-scale blending process (10 L) was performed in a V-blender and a double-cone blender (Yenchen machinery Co., Ltd., Taoyuan, Taiwan) to evaluate changes in operating space according to changes in batch size and manufacturing equipment. These blending processes were performed using optimized blending process parameters at the laboratory-scale, and the drug content and CU were evaluated. However, the Carr index was not evaluated in the pilot-scale blending because the control factors did not have a significant effect on the Carr index from the analysis of DoE in the laboratory-scale. The evaluated results were used to compare statistically with those of pilot-scale blending simulation in the study on DEM application. In other words, these results were used to confirm the efficacy and reliability of the developed DEM model during the scale-up process.

#### 2.2.3. Measurement of IQAs

The drug content and CU were determined through an amlodipine assay. Approximately 100 mg of the mixture, corresponding to the amount of one tablet, was weighed and transferred to a 100 mL volumetric flask. The flask was filled with 5 mL of water and stirred for 10 min. Subsequently, approximately 70 mL of a diluted solution (0.005 M sodium hydroxide solution in methanol) was added to the mixture in the volumetric flask and sonicated for 15 min in a sonic bath, Branson 5800 (Branson Ultrasonics Corporation, Danbury, CT, USA). This volumetric flask was kept at room temperature for 30 min and diluted to exactly 100 mL with a diluted solution. The flask was then inverted 10 times, and the solution was filtered with a 0.45 μm nylon syringe filter (Advantec Toyo Ltd., Tokyo, Japan). Approximately 5 mL of the first filtrate was discarded, and 5 mL of the next filtrate was poured into a 50 mL volumetric flask. This flask was diluted to exactly 50 mL using the mobile phase and stirred for 10 min. The drug content in this prepared solution was analyzed using HPLC (Agilent Technologies, Santa Clara, CA, USA), equipped with a 237 nm detector and a ZORBAX Eclipse Plus C18 column (Agilent Technologies, Santa Clara, CA, USA). The mobile phase prepared was a 45:55 volume mixture of ammonium phosphate buffer and methanol, and the ammonium phosphate buffer was adjusted to pH 2.8 with phosphoric acid. The flow rate was set at 1.0 mL/min, and the injection volume was 20 µL. The drug content was calculated by repeating eight times, and the result was presented as the mean value. In addition, the percentage relative standard deviation (RSD) of the drug content was provided as the CU.

The Carr index, which is used as an indicator of the flowability of the powder, was calculated by the following equation:(3)$$\mathrm{Carr}\;\mathrm{Index}=\frac{\rho_{T}-\rho_{B}}{\rho_{T}}\times 100\%,$$
where *ρ_B_* and *ρ_T_* are the bulk density and tap density corresponding to the formulation materials, respectively. The bulk density and tap density for each material were obtained using an MT-1000 instrument (Seishin Enterprise Co., Tokyo, Japan). The bulk density was determined after overfilling a 100 mL cylinder with the material and removing the excess material with a slide glass. The tap density was measured by tapping 200 times per minute until there was no change in the observed height for the material in the cylinder. The measurement of bulk and tap densities were repeated three times for each material.

### 2.3. Application of DEM to Blending Process

DEM is a numerical method capable of describing the dynamic behavior of discrete particles in a system [12]. The behavior of particles is simulated by solving Newton’s second law and the contact-force model at an iterative discrete time step in DEM [34]. Since the introduction of DEM, several theoretical advances have been made and computer hardware has grown drastically [35,36,37]. The pharmaceutical industry, too, has been quick to incorporate DEM in various applications [8,9,10]. In this study, DEM software (EDEM^TM^; DEM Solutions Ltd., Edinburgh, Scotland) was used to simulate the blending process at the laboratory and pilot-scales. The Hertz–Mindlin (no slip) and Hertz–Mindlin + JKR models were used as contact models to obtain the contact forces acting on each particle. The former was applied to the materials, such as amlodipine besylate, SMCC 90, CCM-Na, and PVP K25, while the latter was used for St-Mg, whose particles experience cohesive forces on their surfaces.

#### 2.3.1. Contact Model

##### Hertz–Mindlin (No Slip) Model

The Hertz–Mindlin (no slip) model is a nonlinear elastic model based on the Hertz theory for contact in the normal direction and the Mindlin and Deresiewicz theory for contact in the tangential direction [38,39]. In this study, the Hertz–Mindlin model was applied to calculate particle–particle and particle–geometry contact forces for amlodipine formulation materials, excluding St-Mg.

In the Hertz–Mindlin (no slip) model, the normal contact force (*F_n_*) between two particles (particle a and particle b) can be stated as a function of the normal overlap (*δ_n_*) [40], which is defined as given below:(4)$$F_{n}=\frac{4}{3}E^{*}\sqrt{R^{*}}\delta_{n}^{2/3},$$
where *E^*^* denotes the equivalent Young’s modulus and *R^*^* denotes the equivalent radius. In addition, the tangential contact force (*F_t_*) between two particles can be expressed as a function of the tangential overlap (*δ_t_*) and tangential stiffness (*S_t_*), which is expressed as given below:(5)$$F_{t}=-S_{t}\delta_{t}.$$

In addition, the Hertz–Mindlin (no slip) model has damping components of normal and tangential forces to describe the effect of energy dissipation [41]. The normal damping force is given by:(6)$$F_{n}^{d}=-2\sqrt{\frac{5}{6}}\beta \sqrt{S_{n}m^{*}}v_{n}^{\vec{rel}},$$
where *m^*^*, *β*, and *S_n_* denote equivalent mass, damping ratio, and normal stiffness, respectively.

The tangential damping force is given by:(7)$$F_{t}^{d}=-2\sqrt{\frac{5}{6}}\beta \sqrt{S_{t}m^{*}}v_{t}^{\vec{rel}},$$
where $v_{t}^{\vec{rel}}$ is the tangential relative velocity between particle a and particle b.

The rolling friction acting on the contact between two particles is accounted for by applying torque to the contacting surfaces [42]:(8)$$T_{p}=-\mu_{r}F_{n}R_{p}\omega_{p},$$
where *T_p_* is the torque on the particle and $\mu_{r}$, *R_p_*, and $\omega_{p}$ denote the rolling friction coefficient, distance from the contact point to the center of mass, and unit angular velocity at the contact point, respectively.

##### Hertz–Mindlin + JKR

A contact model considering adhesion between the particles was developed based on the Hertz theory by Johnson, Kendall, and Roberts [43]. Referred to as the JKR model, this model presents the adhesive contact theory between the stored elastic energy and loss of surface energy [44]. Therefore, the model can be applied to the cohesive system of fine particles or moist materials [42]. In this study, the JKR model was used to describe the particle behavior of St-Mg, whose particles experience cohesive forces on their surfaces.

In the JKR model, the normal elastic contact force (*F_JKR_*), which describes cohesion, can be expressed as a function of contact radius (*a*) and surface energy (γ) [41]:(9)$$F_{JKR}=-4\sqrt{\pi \gamma E^{*}}a^{3/2}+\frac{4E^{*}}{3R^{*}}a^{3}.$$

#### 2.3.2. Calibration of Input Parameters for Amlodipine Formulation

To develop a DEM model that can predict precise particle behaviors during the blending process, input parameters corresponding to each material in the amlodipine formulation should be determined accurately. In general, the input parameters are divided into material and interaction parameters. The material properties include particle size, particle shape, density, Poisson’s ratio, and shear modulus, while the interaction parameters include coefficients of restitution, static friction, and rolling friction [44]. Because all the amlodipine formulation materials used in this study were in the fine powder form (particle size is less than 500 µm), it was difficult to define or directly measure the input parameters, except for particle size, particle shape, and density. Therefore, these input parameters were defined through an indirect method (calibration based on comparison between the simulated and the experimental values). The calibration was performed on the basis of the test that measured a specific bulk property of each material. Subsequently, the experiments were replicated in the DEM simulation, and the input parameters were changed iteratively until the simulation results converged on the experimental results [45]. In this study, the input parameters were defined using calibration based on the bulk tests, such as the static and dynamic angles of repose tests and the basic flow energy (BFE) test.

##### Direct Measurement of Input Parameters

The particle size distribution and diameter (D_10_, D_50_, and D_90_) of each material of the amlodipine formulation was obtained by laser diffraction using the HELOS particle size analyzer (Sympatec GmbH, Clausthal-Zellerfeld, Germany). For each material, approximately 5 g of the sample was taken and used for the measurement. The measurement was repeated thrice for each material.

A scanning electron microscope (SEM) (Model S-4700, Hitachi Co., Tokyo, Japan) was used to define the particle shape of each material. The sample was fixed on the stub using a double-sided adhesive tape and coated with gold under an argon atmosphere in vacuum prior to observation. Microscopic images of the samples at different magnifications were obtained at an acceleration voltage of 20 kV.

The true density of each amlodipine formulation material was determined using AccuPyc II 1340 Pycnometer (Micromeritics Instrument Co., Norcross, GA, USA). The equipment held a sample cell and expansion cell. First, helium gas was injected into the sample cell, and the corresponding pressure was measured. Subsequently, the valve connected to the expansion cell was opened to diffuse the gas towards the expansion cell side. The volume of the sample was calculated from the measured pressure drop and the empty volume in each cell. The true density was calculated by dividing the volume of the calculated sample, i.e., the true density was calculated by dividing the bulk density by the volume of the sample, excluding the empty space between the particles. The measurement of true density was performed thrice for each material.

##### Angle of Repose for Calibration of Input Parameters

The static angle of repose was measured using MT-1000 (Seishin Enterprise Co., Tokyo, Japan). Approximately 40 mg of each material was introduced into the funnel. Subsequently, the material flowed down from the funnel under the influence of gravity (9.8 m/s) to form a stable pile, and the angle of repose for the piled material was measured. The measurements of static angle of repose for each material were repeated five times. The dynamic angle of repose was measured using the rotary drum method [42,46]. The rotary drum was an acrylic cylinder with a diameter of 15 cm. Each material was filled to approximately 30% *v*/*v* of the rotary drum. The drum was mechanically rotated at 25 rpm. The dynamic angle of repose was measured during the rotation, and the measurement for each material was performed five times.

The simulations for static and dynamic angles of repose were performed using EDEM^TM^ software, with the Hertz–Mindlin and Hertz–Mindlin + JKR contact models to calibrate the input parameters, such as material property (e.g., Poisson’s ratio) and interaction parameters between particle–particle and particle–geometry (e.g., coefficient of restitution and coefficient of static and rolling friction). The shear modulus (Pa) for each material was fixed at 10^7^, based on previous studies that showed that the shear modulus with a certain range (10^4^–10^7^) did not affect particle behavior in a significant manner [16,47]. In addition, the particle size of each material was upscaled 100 times so that it did not affect the bulk properties. The particle shape of each material was assumed to be a single sphere to reduce the time required for calibration simulation [5]. In these simulations, the fixed time step and target save interval were set as 30% and 1 s, respectively. For the static angle of repose simulation, a 200 mm diameter flat plate and a funnel with a 60 mm diameter orifice were placed in the simulation domain, as shown in Figure 1a. Then, the static angle of repose for simulation was measured in the same manner as the actual experiment. In addition, a 400 mm diameter rotating drum was constructed in the simulation domain to simulate the dynamic angle of repose, as shown in Figure 1b. The dynamic angle of repose for the simulation was measured in the same manner as the actual experiment. The simulation results of the static and dynamic angles, which were obtained by setting arbitrary input parameters (i.e., Poisson’s ratio, coefficient of restitution, coefficients of static, and rolling friction) values, were compared with the corresponding experimental results. Subsequently, the input parameters were iteratively adjusted to obtain statistically similar simulation results with experimental results for both methods. At this step, the coefficient of static friction and rolling friction, which have a significant effect on the angle of repose, were mainly adjusted [48,49].

##### BFE Test for Calibration of Input Parameters

A Freeman FT4 rheometer (Freeman Technology, Malvern, UK) was used to determine the BFE of each material. BFE refers to the stabilized energy of flow as the major flowability parameter [50]. For the test, a split vessel (25 × 50 mm) and an impeller of 23.5 mm diameter blade were used. This test consisted of conditioning and test cycles. The conditioning cycles were repeatedly performed before the test cycles to remove the packing history of the powder [51]. In the conditioning cycles, the powder was conditioned by moving the impeller downwards and upwards at a helix angle of −5° and rotating the blade clockwise at a tip speed of 40 mm s^−1^. Subsequently, the test cycles were initiated by moving the impeller downwards and upwards at a helix angle of −5° and rotating the blade in the counterclockwise and clockwise directions, respectively. These test cycles were carried out for a total of 11 runs; the first eight runs were carried out with a blade tip speed of 100 mm s^−1^ and additional three runs were performed with blade tip speeds of 70, 40, and 10 mm s^−1^, [52]. The total energy (*E*) was calculated by incorporating the flow energy gradient into the blade depth through the powder bed, as described below [51,53]:(10)$$E=\int_{0}^{h}\left(\frac{T}{R\tan\alpha }+F\right)dh,$$
where *T* is the torque of the impeller blade; *F* is the downward force acting on the blade; *R* is the radius of the blade; *α* is the helix angle of the blade; and *h* is the penetration depth of the blade in the powder bed. Based on Equation (10), BFE was determined by calculating the total energy measured during the downward movement of the impeller blade rotating counterclockwise [53].

The simulation for the BFE test was performed on EDEM^TM^ software with the Hertz–Mindlin and Hertz–Mindlin + JKR contact models, to calibrate the input parameters. The particle size and particle shape for each material were also adjusted to reduce the computational time, as mentioned in the previous section. In addition, the BFE test simulation was performed with a fixed time step and a target save interval of 30% and 0.1 s, respectively. For the BFE test simulation, a cylindrical vessel (25 × 50 mm) and an impeller with a 23.5 mm diameter blade were constructed, as shown in Figure 1c. The particles of each material were generated in the cylindrical vessel under the influence of gravity. BFE test simulations consisting of conditioning and test cycles were performed under the same conditions as the original BFE test. However, the test cycle simulations were performed only in the first eight runs with a blade tip speed of 100 mm s^−1^, unlike the actual BFE test, which was performed for the first eight runs and then for an additional three runs.

#### 2.3.3. Blending Simulation at Laboratory and Pilot-Scales

The blending simulation at laboratory-scale was performed in a V-blender to validate the developed DEM model. The blending simulation at laboratory-scale was conducted with the optimized process parameters derived from the operating space. In addition, the input parameters for each material were set to values defined by either direct measurement or the calibration approach, however, the particle size was upscaled to 100 times the measured D_10_, D_50_, and D_90_ to reduce the computational time required by the blending simulation. For the same purpose, the particle shape was adjusted to a single sphere for all the materials for the amlodipine formulation. Based on these adjustments, the calibration was performed to define input parameters, as described in the previous section. Therefore, it can be concluded that these material properties no longer have effects on bulk behavior in this study [5,54,55]. Based on the determined input parameters, a total of 178,950 particles were generated for the formulation materials to achieve the optimized filling level on a laboratory-scale V-blender. The geometry for the laboratory-scale V-blender used in the simulation was consistent with that used in the actual laboratory-scale experiment, as shown in Table 1. The laboratory-scale blending simulation was performed for 360 s using 24 CPU cores with Hertz–Mindlin (no slip) and Hertz-Mindlin + JKR models. In addition, the time step and target save intervals were set to 25% and 0.5 s, respectively. The blending simulation at laboratory-scale took approximately seven days.

The blending simulations at pilot-scale were performed to evaluate the change of operating space that can arise when batch size or equipment change occurs. For this, a pilot-scale blending simulation was conducted in the V-blender and double-cone blender, with optimized process parameters derived from operating space at the laboratory-scale and the calibrated input parameters. In addition, the particle size and particle shape were adjusted in the same manner as in the blending simulation at laboratory-scale to achieve efficient computational time. To meet the optimized filling level, the total number of particles generated for material formulations was 692,614 in the pilot-scale double-cone blender and 802,240 in the pilot-scale V-blender. The geometries of the V-blender and double-cone blender were consistent with those used in the actual experiments at pilot-scale (Table 1). As in the laboratory-scale blending simulation, the pilot-scale blending simulations were performed for 360 s using 24 CPU cores with Hertz–Mindlin (no slip) and Hertz–Mindlin + JKR models, and the time-step and target save intervals were set to 25% and 0.5 s, respectively. The blending simulations at pilot-scale took approximately 22 days.

## 3. Results and Discussion

### 3.1. Results and Discussion of QbD Application in the Blending Process

#### 3.1.1. Risk Assessment

The IQAs of the mixture, which is the intermediate product for the blending process, were determined based on the effect of homogeneity of mixture directly related to the CQAs of the finished drug product. The IQAs were defined by the drug content, CU, and the Carr index, and they were used in the RA to identify the CPPs for the blending process. As shown in Table 2, the process parameters for the blending process were classified into three risk levels—low (marked in green), medium (marked in yellow), and high (marked in red)—according to their RPN scores for the IQAs describing the homogeneity of mixture. In the blending process, the filling level, rotational speed, and blending time were found to put the IQAs under medium to high risk. These results were determined based on the expertise and experience from previous studies on related processes [56,57,58]. In addition, the Pareto chart was provided on the basis of the RPN scores for each process parameter, as shown in Figure 2. In this chart, the RPN scores for IQAs of the filling level, rotational speed, and blending time were above the threshold value, while the RPN scores for IQAs of the order of input and manufacturing environment were below the threshold value. In addition to this, the Pareto chart, with respect to the filling level, rotational speed, and blending time, showed that the percentage RPN and percentage cumulative RPN for the drug content and CU accounted for 30–40% and 70–80%, respectively. Based on these results, the filling level, rotational speed, and blending time were adopted as CPPs for the careful management, and they were evaluated as control factors with the following Box–Behnken design.

#### 3.1.2. Analysis of DoE Results

A Box–Behnken design was applied in the blending process to quantify the effect of the control factors on the response factors. The filling level (*x_1_*), rotational speed (*x_2_*), and blending time (*x_3_*) were adopted as control factors based on the RA, and the drug content (*y_1_*), CU (*y_2_*), and Carr index (*y_3_*) were set as the response factors. The filling level (*x_1_*) and rotational speed (*x_2_*) examined in this study were in the range 30–70% and 15–25 rpm, respectively, excluding the extremely low or high levels. The DoE for the 15 experiments are listed in Table 3. The analysis of variance (ANOVA) was performed to evaluate the DoE using Design-Expert^®^ software, as listed in Table 4. The results reveal that the models of the drug content and CU were significant (*p* value < 0.05), however, those of the Carr index were not (*p* value > 0.05), although the Carr index satisfied the target values for all the experiments (less than 20). (ANOVA data for the Carr index is not shown). Therefore, the Carr index (*y_3_*) was excluded from subsequent statistical analyses. The control factors and their mutual interactions with significant effects on the responses are presented in the table, but mutual interactions with no significant effects (*p* value > 0.05) are excluded. In addition, the mathematical response surface models for the DoE were generated by the statistical analysis by applying the coded values for each factor.

##### Effect of Control Factors on Drug Content (*y*_1_)

The drug content is an IQA that presents the homogeneity of mixture. It may be directly related to the safety and efficacy of a drug product. Therefore, the drug content should be designed to ensure the desired target value in the amount corresponding to one tablet. The drug contents of the 15 experiments were in the range 90.04–96.77%. The regression analysis of control factors affecting the drug content was used to generate the empirical model in coded terms, as given below:(11)$$y_{1}=+94.11-0.50x_{1}-0.91x_{2}+0.80x_{3}-2.37x_{1}x_{2}+0.42x_{2}x_{3}.$$

All the control factors have a significant effect on the drug content (*p* value < 0.05). In addition, the actual model *R*^2^, adjusted *R*^2^, and predicted *R*^2^ were 0.9829, 0.9733, and 0.9274, respectively (Table 4). It can be concluded that the closeness of these values is suggestive of the goodness of fit. The empirical model demonstrated that the filling level (*x_1_*) and rotational speed (*x_2_*) had negative effects on the drug content, while blending time (*x_3_*) had a positive effect on the same. The mutual interaction of *x_1_* and *x_2_* decreased the drug content, while that of *x_2_* and *x_3_* increased the same. Based on Equation (11), the two-dimensional contour plot was generated for the *x_1_* and *x_2_*, which had the most significant effect on the drug content, as shown in Figure 3a. The contour plot shows that the drug content increases at a lower filling level and faster rotational speed, or at a higher filling level and slower rotational speed. These results suggest that the difference in dynamic behavior of the powder bed in the V-blender occurs according to the filling level and rotational speed [59,60]. The lower filling level and higher rotational speed allow the powder bed to rotate faster in the blender, thereby increasing the drug content [60]. In addition, the increase in drug content at higher filling levels and lower rotational speeds is not fully understood; however, it can be inferred that these conditions have a positive effect on the drug content by blending not only the surface region but also the central region of the powder bed [28].

##### Effect of Control Factors on CU (*y*_2_)

Undesirable CU has a significant effect on the safety and efficacy of the drug product—thus, CU is an IQA. The CU was evaluated as the percentage of RSD by calculating the drug content through an assay. The CU of the 15 experiments was in the range 1.67–9.01%. The regression analysis of a control factor affecting the CU was used to generate the empirical model in coded terms, as given below:(12)$$y_{2}=+5.20+1.06x_{1}-0.07x_{2}-0.60x_{3}+1.13x_{1}x_{3}-3.13x_{2}x_{3}.$$

The filling level (*x_1_*) and blending time (*x_3_*) also have a significant effect on the CU (*p* value < 0.05), while the rotational speed (*x_2_*) is confirmed to have no significant effect (*p* value > 0.05). In addition, the actual model *R*^2^, adjusted *R*^2^, and predicted *R*^2^ were 0.9597, 0.9435, and 0.9018, respectively (Table 4). It can be concluded that the closeness of these values is suggestive of desirability of fit. The empirical model suggests that the *x_1_* has a positive effect on the CU, while the blending time has an *x_3_* on the same. The mutual interaction between *x_1_* and *x_3_* has a positive effect on the CU, while that between *x_2_* and *x_3_* has a negative effect on the same. The relationship between the control factors and the CU is presented as the two-dimensional contour plot for the *x_1_* and *x_3_* that have been identified to have a significant effect on the CU, based on Equation (12), as shown in Figure 3b. The CU decreased at a lower filling level and longer blending time, because they allow the blending of the whole area, including the surface and central regions of the powder bed, as suggested by previous studies [56,60]. Therefore, a low filling level and long blending time are recommended to ensure the CU.

##### Optimization for Blending Process Parameters

To quantify the effect of control factors on the response factors, statistically validated model equations were generated. The process parameters were then optimized to satisfy the target values of the response factors. The blending process for the amlodipine formulation was constrained by the following factors: maximum drug content with the lower (≥95.00%) and upper (≤105.00%) limits; minimum CU within the upper (≤5.00%) limits. In addition, a 95% confidence interval of the optimal condition was used for the control strategy. Based on these conditions, the responses were combined to generate an overall optimal region. The optimal regions of the blending process for the filling level and rotational speed for various blending times, 9 min, 16.5 min, and 24 min, are shown in Figure 4. The yellow space indicates that the desired response values were accomplished simultaneously.

A Monte Carlo simulation was used with MODDE^®^ software to present the robust operating spaces. As shown in Figure 5, robust operating spaces with less than 1% probability of failure are identified as the green space. The operating spaces at blending times 9 min and 16.5 min might be too narrow to obtain a reliable result. Therefore, the optimal settings were derived from the operating space for a blending time of 24 min. These settings are listed in Table 5 and validated with the experimental results for the response factors. Based on the validation results, the absolute biases between the optimal solutions and experimental results were calculated, and it was confirmed that these results were in good agreement, with a slight difference for absolute biases (within 5.0 of absolute biases).

### 3.2. Results and Discussion of DEM Application to Blending Process

A DEM model was developed for the blending process of the amlodipine formulation using EDEM^TM^ software. The input parameters for the DEM model were obtained by the direct measurement and/or calibration approach. In addition, the Hertz–Mindlin and Hertz–Mindlin + JKR contact models were applied to calculate the particle behavior parameters during the blending simulation. The developed DEM model was validated using statistical comparisons with the simulated and experimental results obtained at laboratory-scale. The validated DEM model was then applied to the blending simulations at the pilot-scale to evaluate the change of operating space at the laboratory-scale, derived from the QbD approach. Subsequently, the efficacy and reliability of the validated DEM model was confirmed during the scale-up by statistically comparing the simulation results with the corresponding experimental results at the pilot-scale. In addition, the dynamic behavior of the particles of amlodipine besylate was investigated at laboratory and pilot-scale during the blending simulation.

#### 3.2.1. Definition of Input Parameters

The input parameters of each material were defined to develop a suitable DEM model. Among the input parameters, the particle size, shape, and density were obtained by direct measurement, while Poisson’s ratio, coefficient of restitution, and coefficients of static and rolling friction were determined by the calibration approach. It may not be desirable to define the input parameters by basing the calibration approach only on one bulk test, as the bulk properties of each material are represented by a combination of two or more input parameters [45]. Therefore, the calibration approach was conducted via three types of bulk test—static angle of repose, dynamic angle of repose, and BFE.

The three tests were performed by iteratively adjusting the input parameters in the simulations of three tests until the simulation results were in agreement with the experimental results (within 10.0% of relative biases). Through the calibration approach, the obtained experimental and simulation results for each bulk test are listed in Table 6. Amlodipine besylate and St-Mg were shown to have high static and dynamic angles of repose as compared with other materials, which may have resulted from their poor flowability. To achieve a high angle of repose, the coefficients of static and rolling frictions of these materials were set higher than those of the other materials in the simulations. This is because high coefficients of static and rolling frictions offer a large resistance to linear motion and rotational motion of a particle, which leads to poor flowability of the materials [61,62]. It was also reported that the high coefficients of static and rolling frictions contribute positively to the increase of the static and dynamic angles of repose [63,64]. In addition, the simulation results of the BFE test were obtained in the BFE test simulation, which was conducted with the conditioning cycle and test cycle. The conditioning cycle was performed after the completion of one test cycle to prevent the change in particle arrangement due to the size distribution in each material, as shown in Figure 6. The figure consists of examples of the tests performed on the particles of amlodipine besylate. The green, blue, and red particles in the figure denote D_10_, D_50_, and D_90_, respectively. In the initial state, the particles were distributed uniformly (Figure 6a), however, the segregation of particles according to the size distribution can be observed after one test cycle (Figure 6b). This particle segregation is mitigated by performing a conditioning cycle (Figure 6c). The torque and force acting on the impeller blade during the test cycle in the BFE test simulation are shown in Figure 7a,b. The flow energy gradient obtained based on torque and force is shown in Figure 7c. It can be observed that each amlodipine formulation material, except for St-Mg, exhibits a significant resistance. St-Mg particles exhibit low torque and force, which may be a result of their low bulk density (0.20 g/mL) and poor flowability. This is because low bulk density and poor flowability reduce the flow region of the low stiffness material around the impeller blade, thereby reducing the energy required by that blade [65]. The BFE for each amlodipine formulation material was calculated by substituting the torque and force obtained from the BFE simulation into Equation (10) (Table 6). Consequently, the absolute and relative biases for each bulk test are similar to one another, within a tolerance of approximately 10%. Based on these results, the interaction parameters for particle–particle and particle–geometry, as well as material properties for respective formulation materials, were determined and are listed in Table 7. Furthermore, the material properties of the geometry made entirely of stainless steel are presented based on previous studies [52,66].

#### 3.2.2. Validation of Developed DEM Model

The blending simulation at the laboratory-scale was performed to validate the developed DEM model with the calibrated input parameters. Derived from laboratory-scale operating space, the optimal setting (i.e., filling level: 32% and rotational speed: 24 rpm) was used for the process parameters for the blending simulation. However, the optimal setting of the blending time was not considered because the blending simulation time differs from the physical time of blending experiments in that it is governed by multiple factors, including the number of particles and their shape and size, as well as material properties [67]. Therefore, the comparison between the blending experiment and simulation was performed using the results obtained for their respective times.

The results of the laboratory-scale blending simulation were statistically compared with those of the laboratory-scale blending experiments using Minitab^®^ software (Version 16; Minitab Inc., State College, PA, USA) to validate the efficacy and reliability of the developed DEM model. Both the simulation and experimental results were evaluated for drug content and CU. To obtain these results, sampling cylinder bins were produced in the same region as the region, where the sampling was performed in the actual blending experiment, within the simulation domain as shown in Figure 8a. The drug content was determined by calculating the amlodipine besylate mass (g) in the total mass (g) corresponding to each sampling cylinder bin, and CU was obtained as percentage of RSD with respect to the drug content. In addition, the drug content and CU for the blending experiment were determined as described in Section 2.2.3. Based on these results, a regression analysis was performed for each result, to confirm the statistical similarity between the blending simulation and the blending experiment at laboratory-scale. The regression analysis was conducted at five data points that represented the results (i.e., drug content and CU) obtained for the respective time of experiments and simulations, however, the initial points for the blending simulation and experiment were excluded, as they exhibited large variability in terms of drug content and CU. The regression analysis result is shown in Figure 9a as the fitted line plot for the drug content. The regression model was significant (*p* value < 0.05), and the standard deviation (*S*), *R*^2^, and adjusted *R*^2^ were 3.57, 93.7%, and 91.5%, respectively. The fitted line plot for the CU is shown in Figure 9b. Its regression model was significant (*p* value < 0.05), and the *S*, *R*^2^, and adjusted *R*^2^ were 1.49, 94.3%, and 92.4%, respectively. It can be also observed that data for each result follow the regression closely in both the fitted line plots. Therefore, the regression models for each result suggested the desirability of fit. Based on the regression analysis, it can be concluded that the developed DEM model was validated as effective and reliable because the simulation results were statistically similar to the experimental results.

#### 3.2.3. Evaluation of Change of Operating Space through Developed DEM Model

The blending simulations at the pilot-scale were conducted in the V-blender and double-cone blender to confirm the change of operating space during the scale-up, including the batch size and manufacturing equipment changes. The blending simulations for each blender were performed at the optimal settings of the process parameters (i.e., filling level: 32% and rotational speed: 24 rpm) derived from the laboratory-scale operating space.

The results of each pilot-scale blending simulation (i.e., drug content and CU) were evaluated and compared with those of the laboratory-scale blending simulation to numerically confirm the change of operating space. The drug content and CU of each pilot-scale blending simulation were determined from the sampling cylinder bins, as shown in Figure 8b,c. The drug content and CU in the laboratory-scale blending simulation were obtained using the results from the previous section. To intuitively confirm the change of operating space, the drug content and CU for each scale were presented according to T_index_ (T_i_), as shown in Figure 10. T_i_ denotes the order of time for the blending simulation, which was the same as the time interval for each blending simulation. The change of operating space was confirmed by comparing the T_i_ values of each blending simulation that satisfied the target values of the drug content and CU. The T_i_ values that satisfied the target values for each blending simulation are shown in Figure 11. Furthermore, the blending time (min) that met the target values in the laboratory-scale blending experiments is presented in the figure. The T_i_ value that satisfied the target value was identified as 7 in the laboratory-scale blending simulation, however, the T_i_ values were confirmed as 4 in both the pilot-scale blending simulations. Therefore, it can be concluded that the change of operating space, including batch size and manufacturing equipment changes, derived from the laboratory-scale simulation was predicted in the developed DEM model during the scale-up process. In addition, the results for each pilot-scale blending simulation were compared with the corresponding blending experiment to verify the efficacy and reliability of the developed DEM model that predicts the change of operating space. The blending experiments were conducted in a V-blender and double-cone blender at the same optimal settings as the blending simulations. As a result of the pilot-scale blending experiments, the target values were achieved in 15 min for both blenders. These results differ from the 24 min presented in the laboratory-scale operating space as the optimal setting. Furthermore, this advance in blending time during the scale-up process was consistent with the advance in T_i_ values predicted in the DEM model according to the scale-up. Based on these results, a change of operating space has occurred during the scale-up process predicted in the developed DEM model.

The regression analysis was performed on Minitab^®^ software, to confirm the efficacy and reliability of the scale-up of the DEM model. The analysis was conducted on simulation results and experimental results at the pilot-scale, as described in Section 3.2.2. The results were evaluated for drug content and CU in both the blenders. For the drug content in the V-blender, the fitted line plot is presented in Figure 12a. The regression model was significant (*p* value < 0.05), and its *S*, *R*^2^, and adjusted *R*^2^ were 0.86, 95.9%, and 94.5%, respectively. The fitted line plot of the CU in the V-blender is shown in Figure 12b. The regression model was also significant (*p* value < 0.05), and *S*, *R*^2^, and adjusted *R*^2^ were 0.73, 91.7%, and 88.9%, respectively. These values for each result of the model in the V-blender suggest the goodness-of-fit. In addition, the fitted line plots for the drug content and CU in the double-cone blender are presented in Figure 12c,d, respectively. For the drug content in the double-cone blender, the regression model was significant (*p* value < 0.05), and *S*, *R*^2^, and adjusted *R*^2^ were 1.27, 86.9%, and 82.5%, respectively. The regression model for the CU was also significant (*p* value < 0.05), and *S*, *R*^2^, and adjusted *R*^2^ were 0.70, 93.0%, and 90.6%, respectively. The closeness of these values can represent the goodness-of-fit. Based on these results, it can be concluded that the similarity between the blending simulation results and those of the experiment is assured at the pilot-scale. Therefore, the developed DEM model is effective and reliable for the scale-up process, including changes in batch size and manufacturing equipment.

#### 3.2.4. Comparison of Blending Behavior at Laboratory- and Pilot-Scale

To confirm the blending behavior of the amlodipine formulation at the two scales, snapshots of the blending simulation are presented in Table 8 and obtained against the T_i_ value. The particles of amlodipine besylate are marked in red and other particles for each material are expressed in gold to clearly track the homogeneity of mixture in the blending simulation. In the laboratory-scale blending simulation, the particles of amlodipine besylate were clustered in the surface region on the edge of the powder bed at a T_i_ value of 1. As the blending simulation progressed, the particles of amlodipine besylate are distributed in the central region and are evenly blended at the top and bottom of the powder bed. Furthermore, the particles present a uniformly blended state as a whole for a T_i_ value of 7, which meets the target values of the drug content and CU at the laboratory-scale.

At the initial stage of the pilot-scale blending simulation, the particles were concentrated in the edge region and the middle region of the powder bed in the V-blender and double-cone blender, respectively. These particles gradually distributed in the top and bottom regions of the powder bed. At a T_i_ value of 4, homogeneity of mixture was achieved simultaneously in both the blenders; the amlodipine besylate particles were evenly distributed in the whole of the powder bed. This indicates that the amlodipine besylate particles were distributed more rapidly and evenly in the V-blender at the pilot-scale than at the laboratory-scale. This is because the pilot-scale V-blender and double-cone blender have a longer and wider symmetrical space than the laboratory-scale blenders, which increases the mobility of the powder bed across the plane of symmetry during the blending process [56,68,69]. Based on these observations, it can be concluded that the developed DEM model is useful for confirming the blending behavior of particles on a macroscopic level, which is difficult to obtain experimentally.

## 4. Conclusions

An integrated approach for QbD and DEM has been applied in the amlodipine blending process for the development of a desirable scale-up strategy for pharmaceutical drug manufacturing. The DEM model was developed using the operating space derived from the QbD approach at laboratory-scale. For the development of the DEM model, the input parameters were defined using the calibration approach. The developed DEM model was then validated by comparing the blending simulation results and experimental results at laboratory-scale. The validated DEM model was used to simulate the pilot-scale blending process to evaluate the change of operating space during scale-up. The blending simulations at pilot-scale were performed in a V-blender and double-cone blender. The change of operating space was confirmed by comparing the T_i_ values that satisfied the target values of drug content and CU for laboratory-scale and pilot-scale blending simulations. Also, the actual experiments at the pilot-scale were performed to determine the efficacy and reliability of the developed DEM model during the scale-up process. As a result, the statistical similarities between the pilot-scale experiments and corresponding simulations were confirmed. Therefore, it was concluded that the developed DEM model had good predictability during the scale-up process. Based on these results, this study concludes that the development of the DEM model within the QbD concept can be a useful tool for devising a scale-up strategy for the blending process. In addition, this study provides the basis for further studies regarding the scale-up strategy in other pharmaceutical manufacturing processes.

## Figures and Tables

**Figure 1 pharmaceutics-11-00264-f001:**
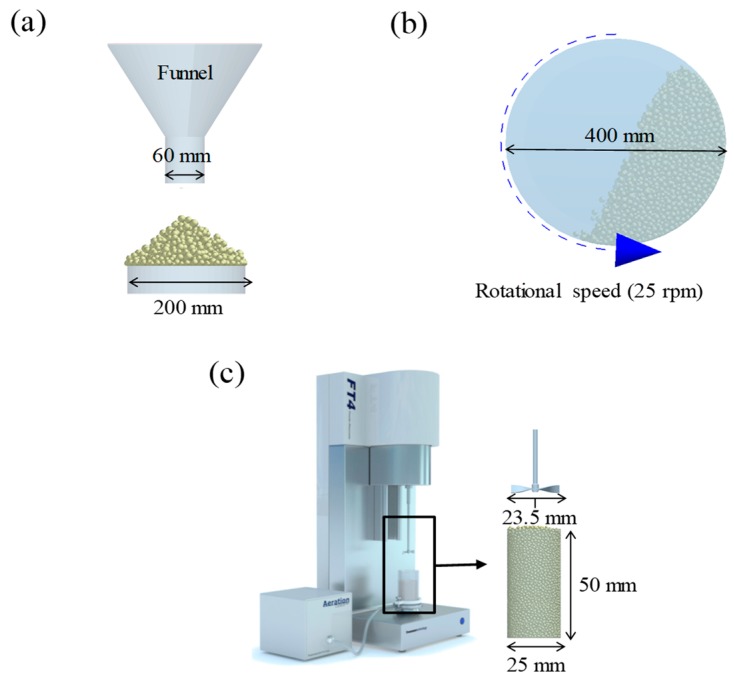
Illustration of geometry for calibration approach: (**a**) static angle of repose, (**b**) dynamic angle of repose (the blue arrow indicates the direction in which the rotating drum revolved), and (**c**) FT4 rheometer.

**Figure 2 pharmaceutics-11-00264-f002:**
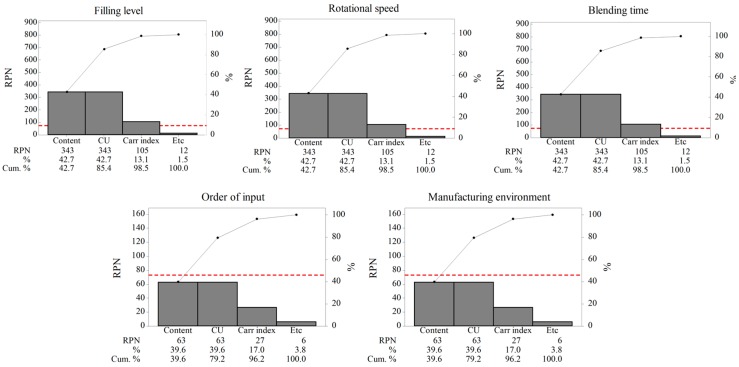
RPN-score-based Pareto chart for each process parameter in the blending process. The red dotted line denotes the threshold value (72.9) and Cum. represents percentage cumulative RPN.

**Figure 3 pharmaceutics-11-00264-f003:**
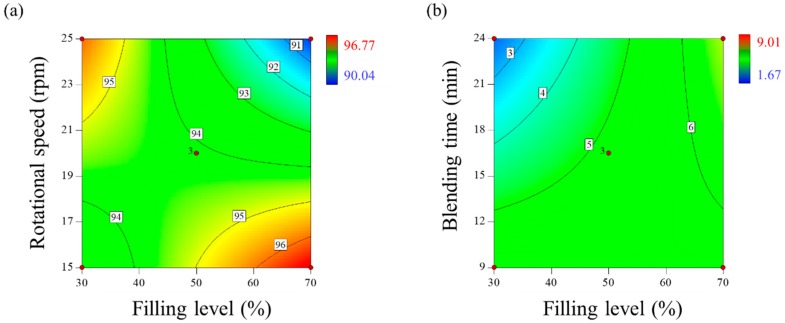
Two-dimensional contour plots to optimize process parameters for blending of amlodipine formulation: (**a**) drug content (%) and (**b**) content uniformity (CU) (%).

**Figure 4 pharmaceutics-11-00264-f004:**
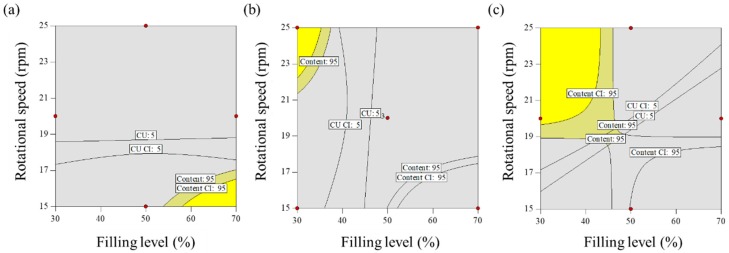
Design spaces to optimize process parameters of blending for amlodipine formulation: blending time at (**a**) 9 min, (**b**) 16.5 min, and (**c**) 24 min. The yellow spaces are parts that satisfy all the target values of IQAs.

**Figure 5 pharmaceutics-11-00264-f005:**
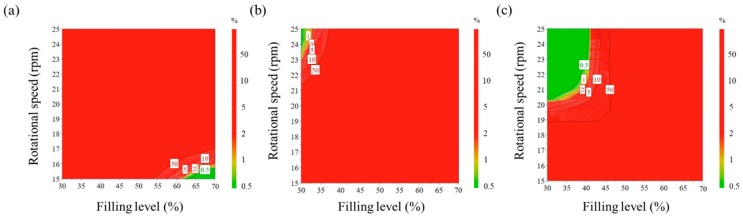
Operating spaces derived by applying Monte Carlo simulation to corresponding design spaces: blending time at (**a**) 9 min, (**b**) 16.5 min, and (**c**) 24 min. The green spaces denote parts with a failure probability of less than 1% of the target values of IQAs.

**Figure 6 pharmaceutics-11-00264-f006:**
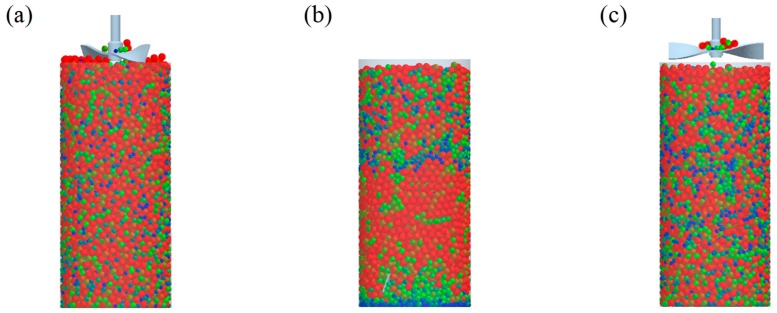
State of particle arrangement for size distribution in the basic flow energy (BFE) test simulation: (**a**) initial state, (**b**) state after the completion of test cycle, and (**c**) state after the completion of conditioning cycle. The green, blue, and red particles are D_10_, D_50_, and D_90_, respectively.

**Figure 7 pharmaceutics-11-00264-f007:**
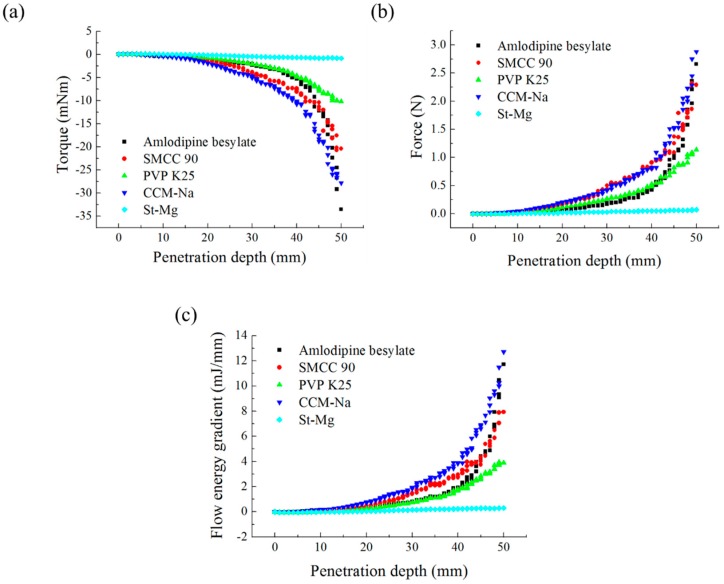
Results of the BFE test simulation: (**a**) torque, (**b**) force, and (**c**) flow energy gradient according to penetration depth acting on impeller.

**Figure 8 pharmaceutics-11-00264-f008:**
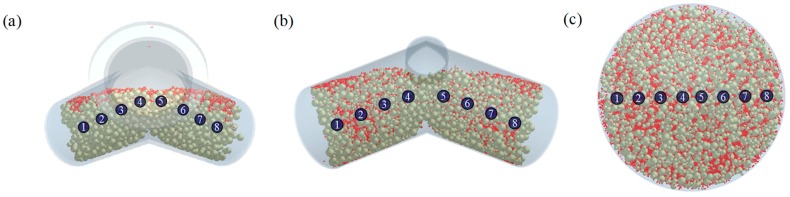
Eight sampling cylinder bins for each blender in the simulation domain: (**a**) V-blender at laboratory-scale, (**b**) V-blender at pilot-scale, and (**c**) double-cone blender at pilot-scale. The view from the normal direction of the powder bed surface.

**Figure 9 pharmaceutics-11-00264-f009:**
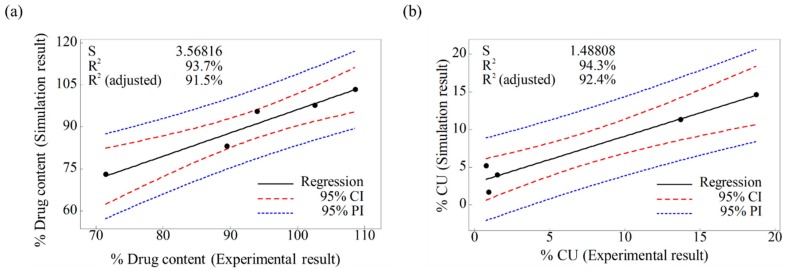
Fitted line plots for the experimental and simulation results at laboratory-scale: (**a**) drug content (%) and (**b**) CU (%). Black line, red dotted line, and blue dotted line denote regression, 95% confidence interval (CI), and 95% prediction interval (PI), respectively.

**Figure 10 pharmaceutics-11-00264-f010:**
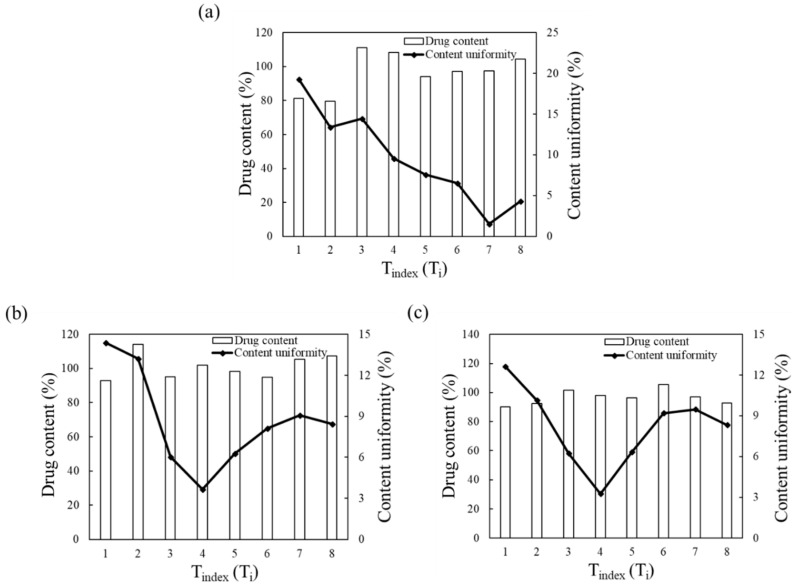
Graphs for each blending simulation result according to T_i_: (**a**) V-blender at laboratory-scale, (**b**) V-blender at pilot-scale, and (**c**) double-cone blender at pilot-scale. For each graph, the bars corresponding to the left axial and the line corresponding to the right axial represent the drug content (%) and CU (%), respectively.

**Figure 11 pharmaceutics-11-00264-f011:**
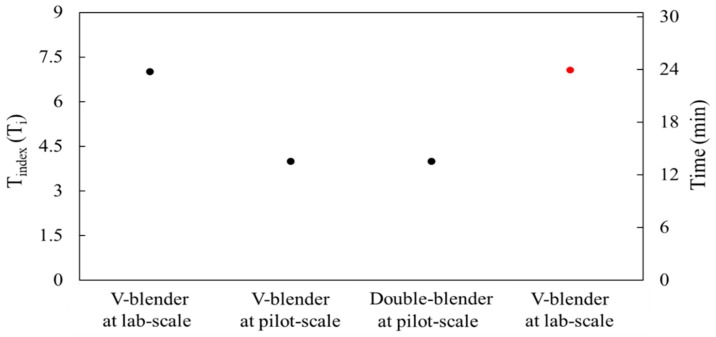
Comparison of T_i_ values and blending time (min) that satisfied the target values for the drug content and CU. The black points corresponding to the left axial are the T_i_ values for each blending simulation and the red point corresponding to the right axial is the blending time for the blending experiment at laboratory-scale.

**Figure 12 pharmaceutics-11-00264-f012:**
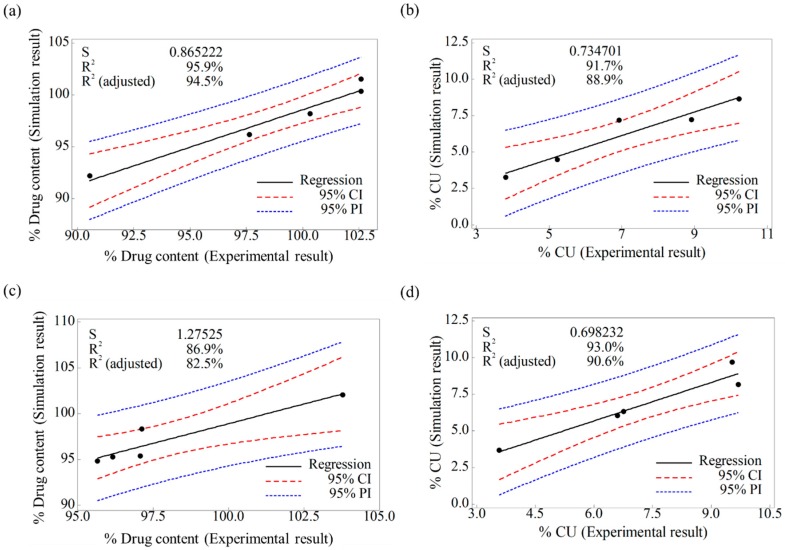
Fitted line plots of the experimental and simulation results at pilot-scale: (**a**) drug content (%) in V-blender, (**b**) CU (%) in V-blender, (**c**) drug content (%) in double-cone blender, and (**d**) CU (%) in double-cone blender. The black line, red dotted line, and blue dotted line denote the regression, 95% confidence interval (CI), and 95% prediction interval (PI), respectively.

**Table 1 pharmaceutics-11-00264-t001:** Blender geometry in actual experiments and simulations.

Blender Geometry	Laboratory-Scale	Pilot-Scale
3 L V-Blender	10 L V-Blender	10 L Double-Cone Blender
Actual geometry	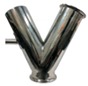	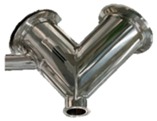	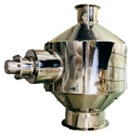
Graphical geometry	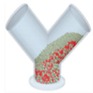	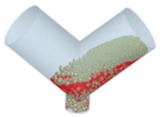	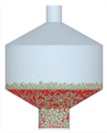

**Table 2 pharmaceutics-11-00264-t002:** Risk assessment of blending process parameters using risk priority number (RPN).

Blending Process Parameters	IQAs	*S*	*P*	*D*	RPN	Risk Level	Justification
Filling level (%)	Carr index	5	7	3	105	Medium	The amount of mixture can have a noticeable effect on the Carr index. Adding too much or too little mixture may result in inhomogeneity of mixture. This may also lead to undesirable flowability. Therefore, the filling level poses a medium risk to the Carr index.
	Drug content	7	7	7	343	High	For desirable blending, it is preferable to add appropriate amounts of the mixture to the volume of blending vessel. If too much or too little mixture is added, the mixture homogeneity cannot be ensured. Therefore, the filling level poses high risk to the drug content.
	CU	7	7	7	343	High	It is desirable to add appropriate amounts of mixture to the volume of the blending vessel. Adding too much or too little mixture may result in inhomogeneity of mixture. This is also directly correlated to the CU. Therefore, the filling level poses a high risk to the CU.
Rotational speed (rpm)	Carr index	5	7	3	105	Medium	Rotational speed can have a noticeable effect on the Carr index. Excessive or sluggish rotational speed can result in inhomogeneity of mixture. This may also lead to undesirable flowability. Therefore, the rotational speed poses a medium risk to the Carr index.
	Drug content	7	7	7	343	High	Rotational speed is directly related to homogeneity of mixture, which should be ensured in the blending process. This homogeneity is explained by the drug content. Therefore, rotational speed poses a high risk to the drug content.
	CU	7	7	7	343	High	Rotational speed is directly related to homogeneity of mixture, which should be ensured in the blending process. Therefore, rotational speed poses a high risk to the CU.
Order of input	Carr index	3	3	3	27	Low	The order of input for the API and excipients, excluding the lubricant, is not a critical process parameter in the blending process, as its effect on the IQAs is limited.
	Drug content	3	3	7	63	Low	
	CU	3	3	7	63	Low	
Blending time (min)	Carr index	5	7	3	105	Medium	The blending time can have a noticeable effect on the Carr index. Too long or too short blending time may lead to undesirable flowability. Therefore, the blending time poses a medium risk to the Carr index.
	Drug content	7	7	7	343	High	Too long or short blending time can result in inhomogeneity of mixture. Therefore, the blending time poses high risk to the drug content.
	CU	7	7	7	343	High	For desirable homogeneity of mixture, appropriate blending time is required. Therefore, blending time poses a high risk to the CU.
Manufacturing environment	Carr index	3	3	3	27	Low	The manufacturing environment, including temperature and humidity, may affect the IQAs. However, the manufacturing environment is kept at constant temperature and humidity. Therefore, this process parameter has negligible effect on the IQAs.
	Drug content	3	3	7	63	Low	
	CU	3	3	7	63	Low	

**Table 3 pharmaceutics-11-00264-t003:** DoE (design of experiments) of blending process parameters with three control factors, and values of response factor for each run order.

Run Order	Control Factors			Response Factors		
	Filling Level (%)	Rotational Speed (rpm)	Blending Time (min)	Drug Content (%)	CU (%)	Carr Index
	*x_1_*	*x_2_*	*x_3_*	*y_1_*	*y_2_*	*y_3_*
1	70	15	16.5	96.77	6.90	19.63
2	70	20	9	92.86	6.19	18.03
3	50	25	9	92.21	9.01	18.50
4	30	20	24	95.70	2.58	18.54
5	50	25	24	94.56	1.67	19.00
6	70	20	24	94.41	7.13	18.52
7	70	25	16.5	90.04	5.18	18.14
8	50	15	9	94.70	2.26	18.84
9	50	15	24	95.36	7.45	18.92
10	50	20	16.5	94.13	5.08	18.40
11	30	25	16.5	95.62	4.18	18.00
12	50	20	16.5	94.42	5.14	18.03
13	30	20	9	93.86	6.15	19.01
14	50	20	16.5	94.13	5.05	19.58
15	30	15	16.5	92.87	4.02	17.94

**Table 4 pharmaceutics-11-00264-t004:** ANOVA (analysis of variance) of responses of experimental design for blending process parameters.

Response	Source	Sum of Squares ^3^	DF ^4^	Mean Square ^5^	*F* Value ^6^	*p* Value	*R* ^2^	Adjusted *R*^2^	Predicted *R*^2^
*y_1_*	Model	36.88	5	7.38	103.25	<0.0001	0.98	0.97	0.93
	*x_1_*	1.97	1	1.97	27.58	0.0005	-	-	-
	*x_2_*	6.61	1	6.1	92.48	<0.0001	-	-	-
	*x_3_*	5.12	1	5.12	71.67	<0.0001	-	-	-
	*x_1×2_*	22.47	1	22.47	314.50	<0.0001	-	-	-
	*x_2_x_3_*	0.71	1	0.71	10.00	0.0115	-	-	-
	Residual ^1^	0.64	9	0.071	-	-	-	-	-
	Cor Total ^2^	37.52	14	-	-	-	-	-	-
*y_2_*	Model	56.16	4	14.04	59.50	<0.0001	0.96	0.94	0.90
	*x_1_*	8.97	1	8.97	38.00	0.0001	-	-	-
	*x_2_*	0.04	1	0.04	0.17	0.6906	-	-	-
	*x_3_*	2.86	2.86	12.10	0.0059	0.0059	-	-	-
	*x_1_x_3_*	5.09	1	5.09	21.55	0.0009	-	-	-
	*x_2_x_3_*	39.25	1	39.25	166.33	<0.0001	-	-	-
	Residual	2.36	10	0.24	-	-	-	-	-
	Cor Total	58.52	14	-	-	-	-	-	-

^1^ Residual: Difference between observed value and predicted value. ^2^ Cor Total: The amount of variation around the mean of the observation. ^3^ Sum of squares: The sum of the squared differences between the overall average and the amount of variation. ^4^ DF (Degree of Freedom): The estimated number of parameters used to calculate the sum of squares of the source. ^5^ Mean square: The sum of squares divided by DF. ^6^
*F* value: Test that compares the mean square of source and the mean square of residual.

**Table 5 pharmaceutics-11-00264-t005:** Optimal settings, optimal solution, and experimental results for responses in operating space.

Optimal Setting			Response Factors					
*x_1_*	*x_2_*	*x_3_*	*y_1_*			*y_2_*		
Filling Level (%)	Rotational Speed (rpm)	Blending Time (min)	Drug Content (%)			CU (%)		
			Optimal Solution	Experimental Results	Absolute Biases	Optimal Solution	Experimental Results	Absolute Biases
32	24	24	96.67	96.82	0.15	0.07	0.98	0.91
40	24	24	95.72	95.53	0.19	0.94	1.54	0.6
36	23	24	95.96	96.25	0.29	1.15	1.88	0.73
32	22	24	96.01	95.70	0.31	1.35	2.14	0.79
40	22	24	95.44	95.51	0.07	2.23	2.75	0.52

**Table 6 pharmaceutics-11-00264-t006:** Experimental results and simulation results of bulk tests for calibration approach.

Materials	Static Angle of Repose			Dynamic Angle of Repose			BFE		
	Experimental Results (°)	Simulation Results (°)	Relative Biases (%)	Experimental Results (°)	Simulation Results (°)	Relative Biases (%)	Experimental Results (mJ)	Simulation Results (mJ)	Relative Biases (%)
Amlodipine besylate	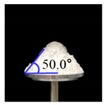	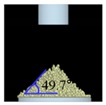	0.6	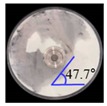	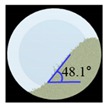	0.8	209.0	195.1	6.6
50.SMCC 90	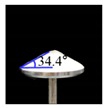	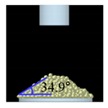	1.5	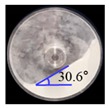	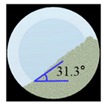	2.3	235.0	228.3	2.8
PVP K25	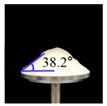	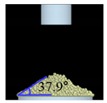	0.8	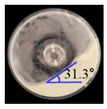	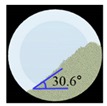	2.2	121.0	129.6	7.1
CCM-Na	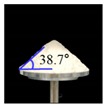	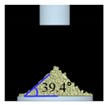	1.8	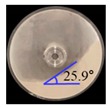	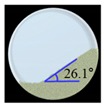	0.8	341.0	330.7	3.0
St-Mg	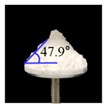	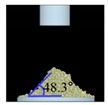	0.8	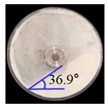	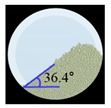	1.4	16.7	17.4	4.2

**Table 7 pharmaceutics-11-00264-t007:** Input parameters defined through direct measurement and calibration approach.

Materials		Input Parameters												
		Material Properties							Interaction Parameters					
		Particle Size (μm)			True Density (g/cm^3^)	Poisson’s Ratio	Shear Modulus (Pa)	Surface Energy (J/m^2^)	Coefficient of Restitution		Coefficient of Static Friction		Coefficient of Rolling Friction	
		D_10_	D_50_	D_90_					P-P ^1^	P-G ^2^	P-P	P-G	P-P	P-G
Amlodipine formulation	Amlodipine besylate	2.74	10.50	28.70	1.36	0.25	10^7^	0.00	0.20	0.20	0.50	0.50	0.50	0.50
	SMCC 90	29.71	106.48	223.88	1.61	0.25	10^7^	0.00	0.30	0.50	0.40	0.40	0.20	0.30
	PVP K25	22.51	62.62	119.75	1.21	0.30	10^7^	0.00	0.30	0.50	0.40	0.40	0.25	0.30
	CCM-Na	23.39	50.48	118.28	1.61	0.30	10^7^	0.00	0.30	0.50	0.40	0.50	0.30	0.30
	St-Mg	1.09	5.52	24.99	1.11	0.30	10^7^	0.02	0.30	0.30	0.50	0.50	0.45	0.40
Geometry	Stainless steel	-	-	-	7.80	0.30	7^10^	-	-	-	-	-	-	-

^1^ P-P: Interaction between particle and particle. ^2^ P-G: Interaction between particle and geometry.

**Table 8 pharmaceutics-11-00264-t008:** Snapshots of blending simulation at each scale according to T_i_ value. Amlodipine besylate particles are marked in red and other particles of each material are marked in gold.

T_i_	Laboratory-Scale (3 L)	Pilot-Scale (10 L)
V-Blender	V-Blender	Double-Cone Blender
1	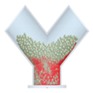	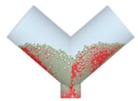	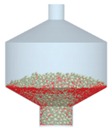
2	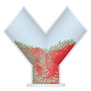	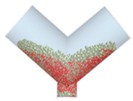	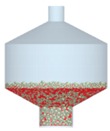
3	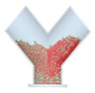	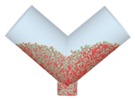	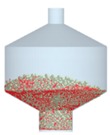
4	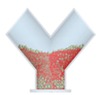	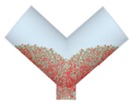	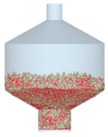
5	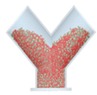	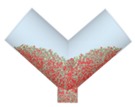	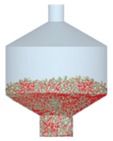
6	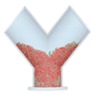	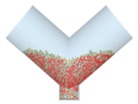	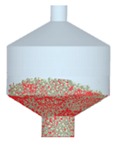
7	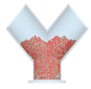	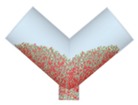	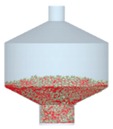

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
