# Peer review of "Scale-Up Strategy in Quality by Design Approach for Pharmaceutical Blending Process with Discrete Element Method Simulation"

_pharmaceutics, 2019, doi:10.3390/pharmaceutics11060264_

Round 1
Reviewer 1 Report
The revised study describes a scale-up strategy of a blending process based on a QbD approach and DEM simulations. Some very interesting facts are described in the paper, i.e. the calculation of a threshold value in the risk analysis and a nice comparison between the laboratory scale and pilot scale blending. In the reviewer´s opinion, the manuscript is well written and the topic is of interest at the present moment, however in order to be suitable for publication, the manuscript would need some revisions.
1. The novelty of the study should be highlighted or expressed more clearly in the final paragraph of the introduction.
2. Line 50 – There are two commas one after another “,,”, if this is a typo, please correct it. Please verify the punctuation marks and spaces throughout the manuscript.
3. Lines 101-102: Please provide some reliable references for the following statement made in this paragraph: “the content uniformity is related to flowability”.
4. Lines 135-136: Provide a comprehensive explanation regarding the calculation and the meaning of the different correlation coefficients.
5. Lines 138-140: We consider that this kind of information belongs rather to the results and discussion section. When mentioning the contour plots you should also provide some information about what is represented in this kind of plots and maybe also depict them as figures.
6. Section 2.2.2. Why did you chose to use two different DoE software, Design Expert for the DoE development and Modde for the Monte Carlo simulation? The Modde software for example, is normally capable of doing both operations. Please provide a pertinent explanation in the manuscript, as well as for the reviser.
7. Line 149-150: A precentral representation would be a better way to present the amlodipine quantitative formulation than the mg/tablet representation used in this section.
8. Lines 406, 431, 447: Try to avoid using bullets (bullet list) when presenting scientific results. It would be more appropriate to use numbered subtitles as it was done in the rest of the manuscript.
9. The Results & Discussion section is well written, however in some paragraphs the discussion is prolonged a little bit too much so that the focus on the essence is lost. Please revise this section and try to shorten the prolonged discussions.
10. Only the main conclusions of the study should be presented in a short section after the Results and Discussion. This conclusions section does not need to recap what has been done in the entire study. Please revise the section and shorten it considerably.
Author Response
The revised study describes a scale-up strategy of a blending process based on a QbD approach and DEM simulations. Some very interesting facts are described in the paper, i.e. the calculation of a threshold value in the risk analysis and a nice comparison between the laboratory scale and pilot scale blending. In the reviewer´s opinion, the manuscript is well written and the topic is of interest at the present moment, however in order to be suitable for publication, the manuscript would need some revisions.
We really appreciate the comments and recommendations that the reviewer provided. The reviewer’s input has been invaluable to the authors during the revision process.
1. The novelty of the study should be highlighted or expressed more clearly in the final paragraph of the introduction.
As suggested, the novelty of this study was inserted in the final paragraph of the introduction. This was highlighted in yellow (Line 85-88). We hope the reviewer considers this point.
2. Line 50 – There are two commas one after another “,,”, if this is a typo, please correct it. Please verify the punctuation marks and spaces throughout the manuscript.
We appreciate the reviewer’s keen point. This was corrected and the change was highlighted in yellow (Line 50).
3. Lines 101-102: Please provide some reliable references for the following statement made in this paragraph: “the content uniformity is related to flowability”.
As suggested, the relevant literature was inserted in the paragraph. The change was highlighted in yellow (Line 104).
4. Lines 135-136: Provide a comprehensive explanation regarding the calculation and the meaning of the different correlation coefficients.
As suggested, the comprehensive explanation regarding the statistical parameters was inserted in the line 141-146. This was highlighted in yellow.
5. Lines 138-140: We consider that this kind of information belongs rather to the results and discussion section. When mentioning the contour plots you should also provide some information about what is represented in this kind of plots and maybe also depict them as figures.
As suggested, the information regarding contour plot was moved into result and discussion paragraph. The change was highlighted in yellow (Line 440-442 and Line 467-468).
6. Section 2.2.2. Why did you choose to use two different DoE software, Design Expert for the DoE development and Modde for the Monte Carlo simulation? The Modde software for example, is normally capable of doing both operations. Please provide a pertinent explanation in the manuscript, as well as for the reviser.
Generally, Design Expert software is used for the design of experiments and statistics analysis of the experiment results. To obtain the design space, the software makes use of an objective function, D(X), called the desirability function to obtain design space. It reflects the desirable ranges for each response (di). The desirable ranges are from zero to one (least to most desirable, respectively). The simultaneous objective function is a geometric mean of all transformed responses as shown below equation:
where m is the number of responses in the measure. If any of the responses or factors fall outside their desirability range, the overall function becomes zero. For simultaneous optimization, each response must have a low and high value assigned to each goal.
However, we wanted to eliminate the probability of failure in the design space derived using the D(X). This can be performed by Monte Carlo simulation. Unfortunately, the software has not the simulation function. Therefore, MODDE software was used to perform Monte Carlo simulation in this study. We hope the reviewer considers this point.
7. Line 149-150: A precentral representation would be a better way to present the amlodipine quantitative formulation than the mg/tablet representation used in this section.
As suggested, the quantitative formulation was modified and this was highlighted in yellow (Line 157-159).
8. Lines 406, 431, 447: Try to avoid using bullets (bullet list) when presenting scientific results. It would be more appropriate to use numbered subtitles as it was done in the rest of the manuscript.
As suggested, the bullet list was modified using the numbered subtitles. The change was highlighted in yellow (Line 429, 455, 473).
9. The Results & Discussion section is well written, however in some paragraphs the discussion is prolonged a little bit too much so that the focus on the essence is lost. Please revise this section and try to shorten the prolonged discussions.
We appreciate the reviewer’s comments. As suggested, we condensed the Results& Discussion section by reducing repetitive expressions and content to avoid the reader’s confusion. We hope the reviewer considers this point.
10. Only the main conclusions of the study should be presented in a short section after the Results and Discussion. This conclusions section does not need to recap what has been done in the entire study. Please revise the section and shorten it considerably.
We appreciate the reviewer’s suggestion and recommendation. The conclusion was condensed to present the main objectives and conclusions of this study more clearly. This change was highlighted in yellow. We hope the reviewer considers this point (Line 702-719).

Reviewer 2 Report
Generally, it is suggested to address the simulation related key information, namely the number of particles per respective simulation, the simulation duration (simulated time and time of simulation), and the used hardware (number of CPU cores) … This information is not of high relevance from a purely pharmaceutical viewpoint, however, since the paper suggests the use of DEM for further work and studies in pharmaceutical scale up processes and manufacturing processes it is recommended to include this information.
In line 50, there are two commas directly after one another.
In line 58 it is said, that FEM is similar to CFD. It is suggested to specify this, e.g. that both methods are located in the field of continuum mechanics.
In line 66 important material attributes are called. Since shape is named as well, it appears that particle attributes are meant here and not just material attributes. Therefore, it is suggested to include the particle size distribution, due to the influence the particle size distribution has on segregation effects.
In line 103 FMEA is called for the first time. It is suggested to provide a short description or write it out in full.
In line 151 the masses of the formulation compounds are given. It is stated, that they are equivalent to the amount of one tablet. It is suggested to state the exact mass of one tablet here.
In line 162 it is stated, that blending processes were performed, …, and drug content and CU were evaluated. Since the Carr Index is already introduced as third response factor, it is suggested to briefly explain why it is excluded here and to refer to the respective text passage, where this is explained in detail.
In line 164 an earlier study is referred to. If possible, please specify a source here.
In line 188 it is stated “… bulk density corresponding to each material …”. It is suggested to write “corresponding to the respective formulations” or “formulation materials”.
In line 192 it is stated that the measurement happens after tapping 200 times per minute. It is suggested to specify the duration of the tapping process.
In line 279 it is stated “… measure the dynamic angle of repose in a stable state of the material …”. It is suggested to refer to an external source for the method of determining the dynamic angle of repose. In many cases the dynamic angle is determined while the drum is rotating.
In line 283 it is stated that the static and dynamic angles of repose were performed to calibrate the input parameters (such as Poisson’s ratio, coefficient of restitution and coefficients of static and rolling friction). It is suggested to describe the process in more detail. Were all 4 parameters varied until a parameter set was found that fitted both experiments? This information would be helpful since there are ways to measure Poisson’s ratio and the coefficient of restitution.
With the calibration procedures the input parameters were calibrated. Was the interaction between the particles and the geometry materials (in the calibration and the blending simulations) addressed as well, additionally to the particle-particle interactions? It is suggested to specify this in the text.
In line 340 it is stated, that the material property is adjusted to such it no longer affects the bulk behaviour. It is suggested to shortly specify this.
In the lines 349 and 350 it is stated “… with optimized process parameters derived from …”. It is suggested to briefly highlight the optimization on lab-scale here, as it was already stated in the lines 159 - 163.
In figure 2 it is suggested to print the threshold value of 72,9 a little more visible and to briefly reference in the text to the passage around line 125, since the value of 72,9 wasn’t stated directly. It is stated indirectly as a result of the given formula (line 125), therefore it is suggested to state 72,9 directly around line 125, so that readers don’t get confused, where the value comes from when looking at figure 2.
In table 4 control factors and their mutual interactions are written. It is suggested to specify, why in case of response factor y1 the mutual interaction x1x3 and in case of the response factor y2 the control factor x2 and the mutual interaction x1x3 aren’t stated.
In line 415 and 416 blending time is assigned to x2 and x3. Based on line 386 x2 should be rotational speed.
In figure 3 a) and b) the response surface plots for the dependencies of x1, x2 and y1 as well as of x1, x3 and y2 are shown, please specify why the other possible dependencies are not shown.
In line 147 it is stated, that 10,000 simulations are performed. In figure 4 different design spaces are shown. Please specify, how the number of simulations is assigned in dependency of the different design spaces, respectively how the design spaces are derived from the number of Monte Carlo simulations.
In figure 5 for y2 there is a deviation between optimal solution and experimental results (especially when compared to the deviation for y1). Please specify how the respective deviations are to be evaluated.
In lines 495 the iterative adjusting with the three bulk tests is mentioned, with the parameter sets shown in table 7 and the consequential simulation results in table 6. Please specify to what extend they were in agreement using the same calibrated variables and whether there was a specific procedure regarding the iterations.
In line 524 it is stated that the material properties of the blender are made entirely of stainless steel. How is the particle-steel interaction addressed in terms of calibration parameters?
In line 544 it is stated that the blending time is not considered. It is suggested to mention to what extend that affects the comparability.
In line 553 it is stated that sampling cylinder bins were produced. It is suggested to mention, that this occurs within the simulation.
In line 560 the 5 data points are explained, in figure 8 there are 8 points per blender. Please specify how the point selection is carried out.
In line 610 it is stated “The results show that the time satisfying target the values of the drug content and CU were confirmed to be 15 min for both the blending experiments.” The grammar sounds unusual and unclear, please check for accuracy.
Author Response
Generally, it is suggested to address the simulation related key information, namely the number of particles per respective simulation, the simulation duration (simulated time and time of simulation), and the used hardware (number of CPU cores) … This information is not of high relevance from a purely pharmaceutical viewpoint, however, since the paper suggests the use of DEM for further work and studies in pharmaceutical scale up processes and manufacturing processes it is recommended to include this information.
We really appreciate the comments and recommendations that the reviewer provided. The reviewer’s input has been invaluable to the authors during the revision process.
The information of the blending simulation condition (the number of particles, number of CPU cores, simulated time and time of simulation) was provided in Section 2.3.3. This was highlighted in yellow (Line 354-356, 360-363, 366-368, 374-377).
In line 50, there are two commas directly after one another.
As suggested, this was corrected and the change was highlighted in yellow (Line 50).
In line 58 it is said, that FEM is similar to CFD. It is suggested to specify this, e.g. that both methods are located in the field of continuum mechanics
We appreciate the suggestion and recommendation of reviewer. The line was modified and the change was highlighted in yellow (Line 58-60).
In line 66 important material attributes are called. Since shape is named as well, it appears that particle attributes are meant here and not just material attributes. Therefore, it is suggested to include the particle size distribution, due to the influence the particle size distribution has on segregation effects.
We appreciate the suggestion of reviewer. The particle size distribution was inserted into the material attributes. This was highlighted in yellow (Line 66-67).
In line 103 FMEA is called for the first time. It is suggested to provide a short description or write it out in full.
As suggested, the full name of FMEA (Failure Mode and Effect Analysis) was provided and this was highlighted in yellow (Line 106-107).
In line 151 the masses of the formulation compounds are given. It is stated, that they are equivalent to the amount of one tablet. It is suggested to state the exact mass of one tablet here.
As suggested, the mass of one tablet was inserted and the quantitative formulation was modified. The change was highlighted in yellow (Line 157-159).
In line 162 it is stated, that blending processes were performed, …, and drug content and CU were evaluated. Since the Carr Index is already introduced as third response factor, it is suggested to briefly explain why it is excluded here and to refer to the respective text passage, where this is explained in detail.
We appreciate the suggestion and recommendation of reviewer. The reason was presented in the paragraph. This was highlighted in yellow (Line 171-173).
In line 164 an earlier study is referred to. If possible, please specify a source here.
We appreciate the reviewer’s suggestion. The expression (“earlier study”) was removed because it seems to confuse the reader. The paragraph was modified and highlighted in yellow (Line 173-174).
In line 188 it is stated “… bulk density corresponding to each material …”. It is suggested to write “corresponding to the respective formulations” or “formulation materials”.
As suggested, “bulk density corresponding to each material” was modified to the “bulk density and tap density corresponding to the formulation materials, respectively.” This modification was highlighted in yellow (Line 197-198).
In line 192 it is stated that the measurement happens after tapping 200 times per minute. It is suggested to specify the duration of the tapping process.
We appreciate the reviewer’s keen point. The duration of the tapping process was specified until there was no change in observed height in the cylinder. This was highlighted in yellow (Line 201-202).
In line 279 it is stated “… measure the dynamic angle of repose in a stable state of the material …”. It is suggested to refer to an external source for the method of determining the dynamic angle of repose. In many cases the dynamic angle is determined while the drum is rotating.
We would like to deliver sincere thanks for the reviewer’s excellent point. The method for measurement of the dynamic angle of repose was modified and highlighted in yellow (Line 289-290). We apologize for any confusion from our mistakes.
In line 283 it is stated that the static and dynamic angles of repose were performed to calibrate the input parameters (such as Poisson’s ratio, coefficient of restitution and coefficients of static and rolling friction). It is suggested to describe the process in more detail. Were all 4 parameters varied until a parameter set was found that fitted both experiments? This information would be helpful since there are ways to measure Poisson’s ratio and the coefficient of restitution.
We appreciate the reviewer’s comment. The input parameters (Poisson’s ratio, coefficient of restitution, coefficient static and rolling friction) were all adjusted until to obtain statistically similar simulation results with experimental results for both methods. At this step, the coefficient of static and rolling friction, which have a considerable effect on the angle of repose, was mainly adjusted. As suggested, the process of the calibration was provided in more detail. This was highlighted in yellow (Line 306-311). We hope the reviewer considers this point.
With the calibration procedures the input parameters were calibrated. Was the interaction between the particles and the geometry materials (in the calibration and the blending simulations) addressed as well, additionally to the particle-particle interactions? It is suggested to specify this in the text.
We appreciate the reviewer’s comment. The calibration was performed to determine the interaction parameters for both particle-particle and particle-geometry, as well as material property. This was specified and highlighted in yellow (Line 291-294).
In line 340 it is stated, that the material property is adjusted to such it no longer affects the bulk behaviour. It is suggested to shortly specify this.
We would like to deliver sincere thanks for the reviewer’s excellent point. The material properties such as particle shape and particle size distribution have an effect on the bulk behavior. However, the material properties used in pharmaceutical manufacturing might be difficult to uniformly define. For example, one material shows a wide variety of particle shapes and sizes. In addition, the number of particles might be impossible to simulate when using the real particle size. To achieve a reasonable simulation time, we used a single sphere for all the materials and the particle size was upscaled to 100 times. The calibration was performed with a single sphere shape and upscaled size. Therefore, the material properties might not influence the bulk behavior in this study. The relevant literature was inserted in the paragraph to help the reader’s understanding (Line 354-356). We hope the reviewer considers this point.
In the lines 349 and 350 it is stated “… with optimized process parameters derived from …”. It is suggested to briefly highlight the optimization on lab-scale here, as it was already stated in the lines 159 - 163.
As suggested, it was briefly mentioned that the optimized process parameters were derived from the operating space at laboratory-scale. The change was highlighted in yellow (Line 367-368).
In figure 2 it is suggested to print the threshold value of 72,9 a little more visible and to briefly reference in the text to the passage around line 125, since the value of 72,9 wasn’t stated directly. It is stated indirectly as a result of the given formula (line 125), therefore it is suggested to state 72,9 directly around line 125, so that readers don’t get confused, where the value comes from when looking at figure 2.
We would like to deliver sincere thanks for the reviewer’s keen point. As suggested, the direct mention of 72.9 was inserted in the formula to prevent confusing the reader and the value was presented in Figure 2 caption (Line 128, 403). We hope the reviewer considers this point.
In table 4 control factors and their mutual interactions are written. It is suggested to specify, why in case of response factor y1 the mutual interaction x1x3 and in case of the response factor y2 the control factor x2 and the mutual interaction x1x3 aren’t stated.
We appreciate the reviewer’s keen point. The source x2 for the y2 response was inserted in Table 4 and this was highlighted in yellow. This was our mistake and we apologize for this point. However, the mutual interactions, which had no significant effect on the response factors, were not presented in Table 4. This was inserted in a paragraph and highlighted in yellow (Line 415-417). These also were not contained in the empirical models to predict response factors.
In line 415 and 416 blending time is assigned to x2 and x3. Based on line 386 x2 should be rotational speed.
We appreciate the reviewer’s keen point. This was corrected and the change was highlighted in yellow (Line 438).
In figure 3 a) and b) the response surface plots for the dependencies of x1, x2 and y1 as well as of x1, x3 and y2 are shown, please specify why the other possible dependencies are not shown.
As suggested, the reason was presented in the paragraph to specify the reviewer’s comment. This was highlighted in yellow (Line 441-442, 467-468). We hope the reviewer considers this point.
In line 147 it is stated, that 10,000 simulations are performed. In figure 4 different design spaces are shown. Please specify, how the number of simulations is assigned in dependency of the different design spaces, respectively how the design spaces are derived from the number of Monte Carlo simulations.
As suggested, the reason was presented in the paragraph to specify the reviewer’s comment. This was As suggested, we provided the Monte Carlo simulation method in more detail. A total of 10,000 simulations were performed using a random number for respective design space. This was highlighted in yellow (Line 153-156). We hope the reviewer considers this point.
In figure 5 for y2 there is a deviation between optimal solution and experimental results (especially when compared to the deviation for y1). Please specify how the respective deviations are to be evaluated.
Is this comment regarding Table 5? We reviewed with Table 5.
We appreciate the reviewer’s comment. The absolute biases between the optimal solution and experimental results were calculated, and it was considered these results were in good agreement with a slight difference for absolute biases (within 5.0). This was highlighted in yellow (Line 488-491).
In lines 495 the iterative adjusting with the three bulk tests is mentioned, with the parameter sets shown in table 7 and the consequential simulation results in table 6. Please specify to what extend they were in agreement using the same calibrated variables and whether there was a specific procedure regarding the iterations.
As suggested, we have specified the extent that they were in good agreement between experimental and simulation results (within 10% of relative biases) (Line 525-527). Also, the procedure for iterative adjustment was inserted in Section 2.3.2. (Specifically, Section 2.3.2.2.). This was highlighted in yellow (Line 306-311).
In line 524 it is stated that the material properties of the blender are made entirely of stainless steel. How is the particle-steel interaction addressed in terms of calibration parameters?
We would like to deliver sincere thanks for the reviewer’s keen point. We apologize for any confusion by missing out the input parameters regarding the particle-geometry. As suggested, the mention of particle-geometry interaction in terms of the calibration was provided and highlighted in yellow (Line 553-556).
In line 544 it is stated that the blending time is not considered. It is suggested to mention to what extend that affects the comparability.
As suggested, we presented the explanation about the blending time in the paragraph. This was highlighted in yellow (Line 579).
In line 553 it is stated that sampling cylinder bins were produced. It is suggested to mention, that this occurs within the simulation.
As suggested, the description (the sampling cylinder bins were generated in the simulation domain) was provided and highlighted in yellow (Line 584-585).
In line 560 the 5 data points are explained, in figure 8 there are 8 points per blender. Please specify how the point selection is carried out.
We apologize for any confusion caused by the ambiguous description of points. The 5 data points denote the results (API content and content uniformity) obtained for the respective time of experiments and simulations. 8 points in figure 8 represent the sampling points in each blender. The explanation regarding the reviewer’s comment was presented to avoid confusion of the reader. This was highlighted in yellow (Line 591-594).
In line 610 it is stated “The results show that the time satisfying target the values of the drug content and CU were confirmed to be 15 min for both the blending experiments.” The grammar sounds unusual and unclear, please check for accuracy.
We would like to deliver sincere thanks for the reviewer’s keen point. The paragraph was modified to avoid confusion of the reader. This was highlighted in yellow (Line 638-639).
